# Positive density dependence acting on mortality can help maintain species-rich communities

**Thomas G Aubier***

Department of Evolutionary Biology and Environmental Studies, University of Zurich, Zurich, Switzerland

**Abstract** Conspecific negative density dependence is ubiquitous and has long been recognized as an important factor favoring the coexistence of competing species at local scale. By contrast, a positive density-dependent growth rate is thought to favor species exclusion by inhibiting the growth of less competitive species. Yet, such conspecific positive density dependence often reduces extrinsic mortality (e.g. reduced predation), which favors species exclusion in the first place. Here, using a combination of analytical derivations and numerical simulations, I show that this form of positive density dependence can favor the existence of equilibrium points characterized by species coexistence. Those equilibria are not globally stable, but allow the maintenance of species-rich communities in multispecies simulations. Therefore, conspecific positive density dependence does not necessarily favor species exclusion. On the contrary, some forms of conspecific positive density dependence may even help maintain species richness in natural communities. These results should stimulate further investigations into the precise mechanisms underlying density dependence.

**\*For correspondence:**
thomas.aubier@normalesup.org

**Competing interests:** The author declares that no competing interests exist.

## Introduction

The tremendous diversity of species in ecological communities has long motivated ecologists to explore how this diversity is maintained (*Hutchinson, 1959*; *Chesson, 2000*; *Hubbell, 2001*; *Levine et al., 2017*). Species richness in a local community is the result of several processes that act at different scales, none of them being mutually exclusive (*Götzenberger et al., 2012*). At regional and global scales, these include randomness and dispersal processes (*Hubbell, 2001*; *Leibold and Chase, 2018*). At local scale, in addition to abiotic factors (physical constraints of the environment), biotic interactions determine community assembly (*Gause, 1934*; *Hutchinson, 1961*; *Holt, 1977*; *Tilman, 1982*; *Chesson, 2000*; *Gross, 2008*). In particular, some species are better competitors than others, and competitive imbalances can lead to the exclusion of less competitive species (*Gause, 1934*; *Levin, 1970*; *Tilman, 1982*; *Meszéna et al., 2006*).

*Interspecific competition for resources* (see *Box 1* for a glossary of terms in italics) has been recognized as one of the main drivers of species exclusion (*Gause, 1934*; *Tilman, 1982*). Additionally, *interspecific reproductive interference* – i.e. any interspecific sexual interaction reducing the reproductive success of females – can inhibit species coexistence. Such interspecific sexual interactions are common in nature, especially among closely related species (*Gröning and Hochkirch, 2008*), and can cause species exclusion more easily than competition for resources, as shown theoretically (*Kuno, 1992*; *Yoshimura and Clark, 1994*; *Kishi and Nakazawa, 2013*; *Schreiber et al., 2019*) and empirically in some species (*Takafuji et al., 1997*; *Kishi et al., 2009*; *Takakura et al., 2009*; *Crowder et al., 2010*; *Crowder et al., 2011*).

In the face of these negative interspecific interactions, many mechanisms favoring species coexistence have been identified. These include niche separation (*Gause, 1934*), predatory/herbivory

## Box 1. Glossary.

### Interactions

**Intraspecific competition for resources –** Any form of competition in which conspecifics (i.e. individuals of the same species) compete for resources.

**Interspecific competition for resources –** Any form of competition in which heterospecifics (i.e. individuals belonging to different species) compete for resources.

**Interspecific reproductive interference –** Any sexual interaction in which heterospecifics reduce female reproductive success (or female function).

### Density dependence

**Conspecific negative density dependence –** Decline in the population growth rate with increasing local density of conspecifics. It typically results from intraspecific competition for resources.

**Heterospecific negative density dependence –** Decline in the population growth rate with increasing local density of heterospecifics. It typically results from interspecific competition for resources.

**Conspecific positive density dependence –** Increase in the population growth rate with increasing local density of conspecifics. It can arise from many different mechanisms (*Figure 1*). This is the focus of this study.

**Heterospecific positive density dependence –** Increase in the population growth rate with increasing local density of heterospecifics. It can arise from any mutualistic interaction (e.g. interspecific facilitation among plant species). This is not the focus of this study.

### Frequency dependence

**Heterospecific negative frequency dependence –** Decline in the population growth rate with increasing local frequency of heterospecifics vs. conspecifics. It typically results from interspecific reproductive interference. This can also be called conspecific positive frequency dependence.

**Heterospecific positive frequency dependence –** Increase in the population growth rate with increasing local frequency of heterospecifics vs. conspecifics. It typically results from the interplay between intraspecific and interspecific competition for resources. This can also be called conspecific negative frequency dependence.

interactions (*Chesson and Huntly, 1997*), positive interactions (*Gross, 2008*), crowding effects (*Gavina et al., 2018*), and individual-level variations (*Uriarte and Menge, 2018*; but see *Hart et al., 2016*). Notably, species often differ in their use of multiple-limiting resources (*Tilman, 1982*), causing species to limit their own population growths more than they limit others (*Adler et al., 2007*). A core tenet of Chesson's coexistence theory – one of the well-developed coexistence theories – is precisely that negative density dependence must be stronger among conspecifics (*conspecific negative density dependence*) than among heterospecifics (*heterospecific negative density dependence*) for species to coexist in two-species systems (*Chesson, 2000*). Even though this criterion does not hold in multispecies communities (*Barabás et al., 2016*; *Song et al., 2019*), the importance of conspecific negative density dependence – when the growth rate of a population decreases as its density increases – for species coexistence in two-species systems is well accepted (*MacArthur, 1970*; *Chesson, 2000*; *McPeek, 2012*). Indeed, conspecific negative density dependence favors the existence of a coexistence equilibrium (i.e. the 'feasibility condition' is fulfilled) that is globally stable (all species can invade even if they are rare initially; i.e. the 'global stability condition' is fulfilled) (*Case, 2000*).

**Conspecific positive density dependence associated with:**

| increased reproduction | reduced mortality | |
|---|---|---|
| | **reduction of the risk of predation** | **reduction of other sources of mortality** |
| • increased probability of finding a mate<br>ex: copepod, insect- or wind-pollinated plants<br>• increased probability for sperm and egg to meet (if external reproduction)<br>ex: starfish, sea urchin<br>• increased success of rearing youngs<br>ex: wild dog, meerkat | • increased vigilance at the group level<br>ex: meerkat, bighorn sheep, baboon<br>• reduced vulnerability of individual prey (dilution)<br>ex: aposematic prey species, mussel, cod, starfish, woodland caribou | • increased foraging efficiency<br>ex: wild dog, black-browed albatross, chimpanzee, wolf, social spider<br>• improved resistance against environmental stress<br>ex: marmot, mussell, grass in arid environment |

**Figure 1.** Mechanisms causing conspecific positive density dependence, and affecting reproduction or mortality. For references and for a more exhaustive list, see the reviews by *Courchamp et al., 1999*; *Stephens et al., 1999*; *Berec et al., 2007*; *Kramer et al., 2009*. Note that two or more mechanisms causing positive or negative density dependence can occur simultaneously (*Berec et al., 2007*).

In many species, however, individuals benefit from the presence of conspecifics, resulting in *conspecific positive density dependence* – i.e. the growth rate of a population increases as its density increases (a phenomenon that is commonly referred to as 'Allee effect' when it occurs at low density). Contrary to conspecific negative density dependence, conspecific positive density dependence can inhibit the coexistence of species interacting negatively with each other (e.g. via competition for resources or reproductive interference) by reducing even further the growth rate of inferior competitors that are at lower density than superior competitors (as shown theoretically by *Wang et al., 1999* and *De Silva and Jang, 2015*). More precisely, the effect of conspecific positive density dependence on coexistence is two-fold. First, it can constrain the conditions under which there is stable coexistence (i.e. the 'feasibility condition' is constrained). Second, it can inhibit the invasion of species that are at low density initially (i.e. the 'global stability condition' is unfulfilled). Therefore, in concert with competition for resources and reproductive interference, conspecific positive density dependence can be a potent mechanism inhibiting the coexistence of competing species. Note that *heterospecific positive density dependence* – when the growth rate of a population increases as the density of heterospecifics increases – has also been documented (it can arise from any mutualistic interaction; for example *Bruno et al., 2003*) and can promote species coexistence (*Gross, 2008*). In the present study, however, I focus exclusively on the implication of conspecific positive density dependence for coexistence.

Conspecific positive density dependence has been described for most major animal taxa (reviewed by *Courchamp et al., 1999*; *Stephens et al., 1999*; *Kramer et al., 2009*), and can be caused by a variety of mechanisms, such as mate limitation (*Gascoigne et al., 2009*), cooperative feeding (*Carbone et al., 1997*), cooperative defense (*Angulo et al., 2018*), predator satiation (*Gascoigne and Lipcius, 2004*) or anti-predator strategies (like aposematism, *Mallet and Joron, 1999*; *Joron and Iwasa, 2005*). In plants, conspecific positive density dependence can be driven by inbreeding depression (*Willi et al., 2005*), pollen limitation (*Sih and Baltus, 1987*; *Groom, 1998*) or substrate modification (favoring seedling establishment, *Bruno et al., 2003*). Interestingly, conspecific positive dependence can associate with increased reproduction or reduced mortality (*Figure 1*; as emphasized by *Berec et al., 2007*). However, theoretical models investigating the determinants of species coexistence have considered conspecific positive density dependence acting on birth

rates (*Wang et al., 1999*) or net growth rates (defined as the difference between birth rate and mortality rate; *De Silva and Jang, 2015*), but not on mortality rates alone. Decreased mortality rates at high population density give rise to conspecific positive density dependence in many species (*Figure 1*; and see the description of specific natural history examples in *Figure 2*), yet the effect of this form of conspecific positive density dependence on species coexistence has not been investigated explicitly.

One might expect that conspecific positive density dependence should inhibit species coexistence whatever the exact underlying mechanism; in all cases, the most competitive species is at high density and therefore benefits the most from conspecific positive density dependence, thereby promoting species exclusion (*Wang et al., 1999*; *De Silva and Jang, 2015*). Nonetheless, density dependence acting on extrinsic mortality (e.g. density-dependent predation) also changes the overall magnitude of extrinsic mortality, yet extrinsic mortality can inhibit species coexistence in the first place (as shown by *Holt, 1985*, using the global stability criterion). For this reason, conspecific positive density dependence acting only on mortality may yield to a different outcome in term of species coexistence.

The primary goal of this analysis is to show that conspecific positive density dependence associated with reduced mortality can help maintain coexistence among competing species. First, I analyze analytically a two-species model with asymmetric resource competition and conspecific positive density dependence acting on mortality, and I show that a locally stable coexistence equilibrium often exists under such positive density dependence. Second, using numerically simulations, I show that this form of conspecific positive density dependence substantially increases the coexistence region in models accounting for other forms of species asymmetry, as differences in basal mortality (in addition to symmetric resource competition) or asymmetric reproductive interference. In other words,

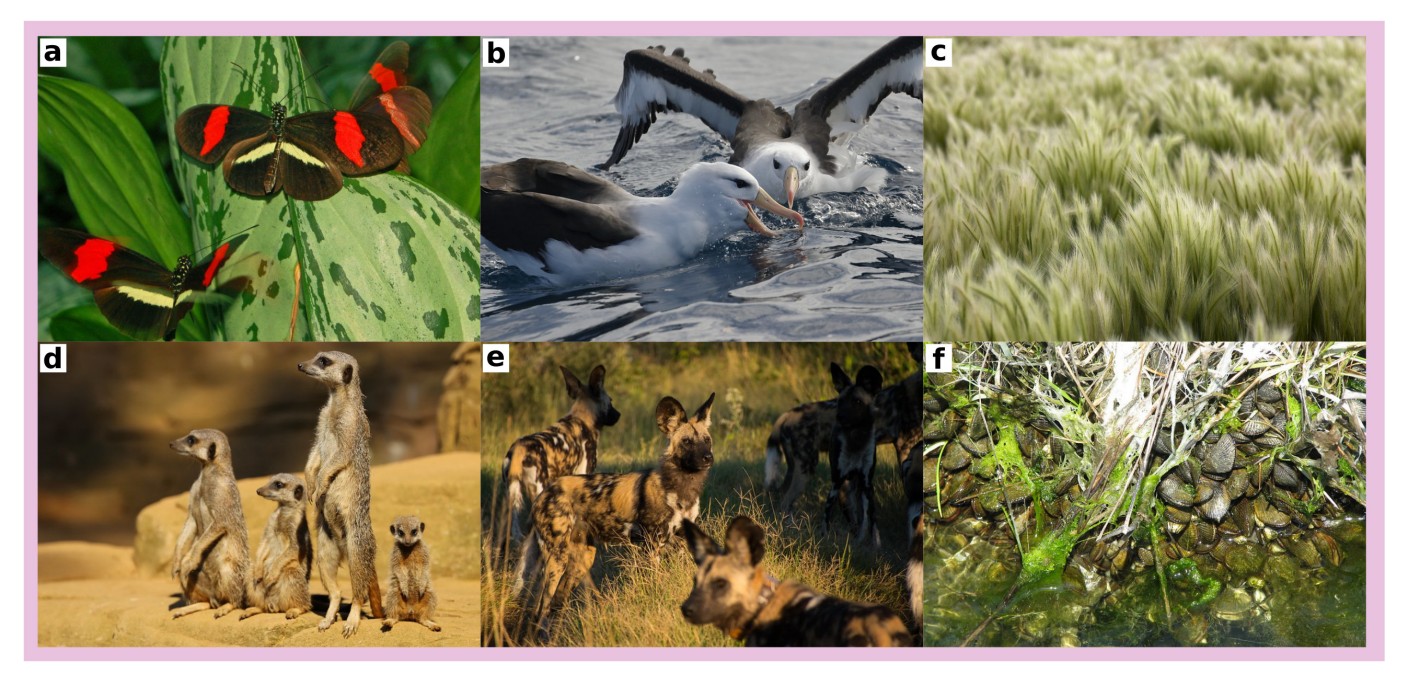

**Figure 2.** Example of organisms undergoing conspecific positive density dependence that associates with reduced mortality. (a) Heliconius butterflies are aposematic (toxic and conspicuous) and therefore benefit from reduced predation when conspecifics are abundant. (b) Black-browed albatrosses in large flocks benefits from increased foraging efficiency and therefore from increased survival (*Grünbaum and Veit, 2003*). (c) Bottlebrush squirreltail in high density have a high survival rate and a high establishment success in arid grasslands (*Sheley and James, 2014*). (d) Meerkats in large group benefit from increased vigilance at the group level, reducing the risk of predation (*Clutton-Brock et al., 1999*). (e) Wild dogs in large group benefit from a high hunting success and defend effectively their kill against kleptoparasitism, thereby increasing their survival (*Fanshawe and Fitzgibbon, 1993*; *Carbone et al., 1997*). (f) Ribbed mussels in high density benefit from reduced crab predation and from improved winter ice resistance (*Bertness and Grosholz, 1985*). Photo credits: (a) Ettore Balocchi; (b) Ed Dunens; (c) Jeffry B. Mitton; (d) Ronnie Macdonald; (e) Barbara Evans; (f) Kerry Wixted.

this form of conspecific positive density dependence generally inhibits the global stability of coexistence but can also increase the feasibility domain of the coexistence equilibrium. Third, using simulations of a multi- (> 2) species model, I then show that conspecific positive density dependence associated with reduced mortality can thereby maintain species-rich communities.

## Model

### Two-species model with asymmetric competition for resources
Model
First, I analyze a simple model accounting for (1) asymmetry in competitive abilities among species and (2) a positive density-dependent mortality term. By being analytically tractable, this model precisely pinpoints the effect of conspecific positive density dependence associated with reduced mortality on species coexistence – i.e. on the feasibility and the global stability of the coexistence equilibrium. All analytical derivations and formal mathematical justifications for the results in this section are detailed in *Supplementary files 1A and 1B*. The computer code of the simulations and of the analyses is provided as *Source code 1* (Python, version 2.7.15).

I consider Lotka-Volterra competition equations, with rescaled variables $t$, $n_1$ and $n_2$ that are dimensionless. Namely, $t$ is time scaled to the growth rate, and $n_1$ and $n_2$ denote the abundances of the two competing species relative to their carrying capacity (scaling is detailed in *Supplementary file 1A*, and follows *Joron and Iwasa, 2005*). Both species are assumed to exhibit conspecific negative density dependence due to competition for resources, such that changes in abundances are chosen as logistic regulation rules. Additionally, the presence of species 1 inhibits the growth of species 2, whereas species 2 has no effect on the growth of species 1 – i.e. there is asymmetric interspecific competition for resources leading to heterospecific negative density dependence in one species only. In addition to local competition for resources, conspecific positive density dependence acting on a mortality term affects species abundances. For instance, this additional term of mortality may approximate the direct effect of predation upon the set of competing species (as assumed by *Holt, 1985*, and *Joron and Iwasa, 2005*). Aside from asymmetric competition, species are assumed to be similar in order to keep the model analytically tractable. In particular, species have the same carrying capacity, and the growth of both species is constrained by the same type of conspecific density dependence (same conspecific negative density dependence driven by competition for resources, and same conspecific positive density dependence acting on mortality). Dynamics are governed by the equations:

$$\begin{cases} \dfrac{dn_1}{dt} = n_1[1 - n_1 - d \times D(n_1)] \\ \dfrac{dn_2}{dt} = n_2[1 - n_2 - \alpha_1 n_1 - d \times D(n_2)] \end{cases} \tag{1a}$$

With the non-linear density-dependent mortality function:

$$D(n_i) = \frac{1}{1 + s\,n_i} \tag{1b}$$

Parameter $\alpha_1 \in [0,1]$ reflects the intensity of asymmetric competition for resources. If $\alpha_1 = 0$, the population dynamics of the two species are independent from each other. If $\alpha_1 > 0$, species 1 benefits from a competitive advantage and inhibits the growth of species 2 (heterospecific negative density dependence). Additionally, negative density dependence is assumed to be stronger among conspecifics than among heterospecifics, hence $\alpha_1 \leq 1$ (following the general coexistence criterion in a two-species system; *Chesson, 2000*). The density-dependence factor $s \geq 0$ corresponds to the decline rate of mortality at $n_i = 0$ ($D'(0) = -s$) (*Mallet and Joron, 1999*; *Joron and Iwasa, 2005*). If $s = 0$, the two species have the same basal density-independent mortality rate $d \in [0,1]$ (relative to the growth rate); this corresponds to the situation without positive density dependence in the system. If $s > 0$, density-dependent mortality is characterized by a hyperbolic decrease with species abundance – the most abundant species has the lowest mortality rate $d \times D(n_i)$ (*Figure 3*). In supplementary analyses, other linear or non-linear mortality functions are implemented (*Figure 3—figure supplement 1*).

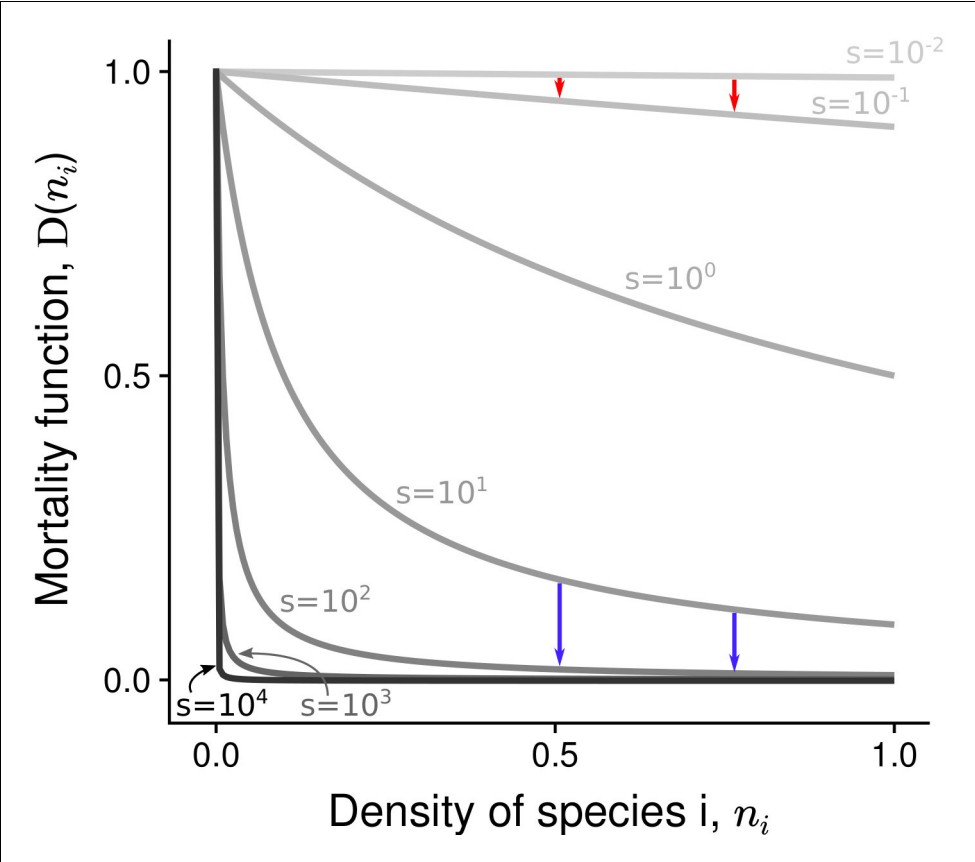

**Figure 3.** Non-linear mortality functions with different density-dependence factors (*s*). Pairs of arrows illustrate the non-linear effect of increased *s* on mortality reduction. Depending on the value of the density-dependence factor *s*, the most abundant species either benefits the most (red arrows) or the least (blue arrows) from increased *s*. The online version of this article includes the following figure supplement(s) for figure 3:

**Figure supplement 1.** Non-linear and linear mortality functions with different density-dependence factors (*s*) tested in supplementary analyses.

**Figure supplement 2.** Sensitivity to increased density-dependence factor.

**Figure supplement 3.** Effect of increased density-dependence factor on the mortality of the least vs. the most abundant species.

For $s>10^2$, mortality is strongly reduced, even at very low density, and there is almost no reduction of mortality over most of the range of density. This extreme situation pinpoints the effects of increased *s*, and the full range of $s>0$ is therefore considered. Remember however that there is a reduction of mortality with increased density at all densities – i.e. conspecific positive density dependence occurs at all densities – only for $s \in [0, 10^2]$ (as emphasized in all graphs).

The 'usual' form of the Lotka-Volterra model from *Equation 1a* can easily be derived (following *Williams and Banyikwa, 1981*). The density-dependent mortality term decreases not only effective intrinsic growth rates, but also effective carrying capacities – i.e. density values reached at equilibrium without competitor (see *Supplementary file 1A*). In particular, increased density-dependence factor (*s*) associates with high effective carrying capacities, by reducing the intensity of the density-dependent mortality term. As a result, the positive density-dependence factor *s* is likely to determine the nature of the equilibrium points of the system.

Note that the population dynamic of species 1 is not affected by the density of species 2; only species 2 can be excluded.

## Analytical resolution

Depending on the parameters considered, this system can be characterized by a single stable equilibrium point where both species coexist (i.e. a 'feasible' equilibrium point of coexistence where both species persist with $n_i>0$, hereafter called 'coexistence equilibrium') (white and gray zones in *Figure 4a–c*; e.g., *Figure 5a* and *Figure 5c*). In particular, under strong asymmetric competition, conspecific positive density dependence characterized by an intermediate factor ($s$) leads to the loss of the coexistence equilibrium (black zone in *Figure 4*; e.g., *Figure 5b*). Above a threshold value (identified analytically in *Supplementary file 1B*, and represented by a dashed red line in *Figure 4*), increased density-dependence factor $s$ therefore favors the existence of the coexistence equilibrium.

The coexistence equilibrium is always locally stable when it exists (*Figure 5*). Moreover, for a low density-dependence factor $s$, the least competitive species can invade when it is rare, just like

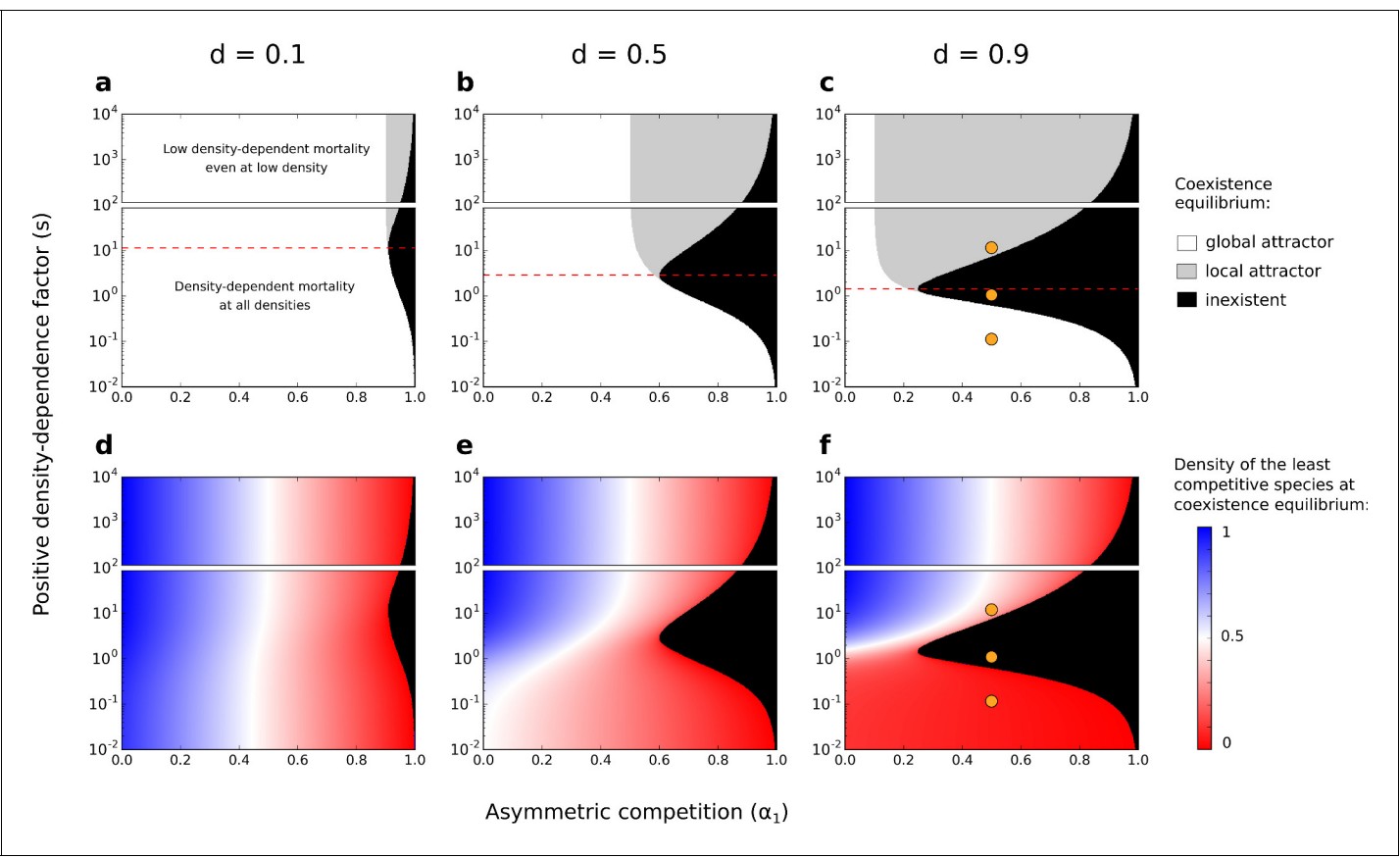

**Figure 4.** Effects of asymmetric competition for resources ($\alpha_1$) and positive density-dependence factor ($s$) on coexistence. Different values of the basal mortality rate ($d$) are also tested. If the coexistence equilibrium exists, it is either a global attractor (it is reached independently of the initial conditions, as long as the two species start at density >0) or a local attractor (it is not reached if the least competitive species is initially at a low density) (**a–c**). In each subfigure, the range of $s$ is arbitrarily divided in two; one range where there is conspecific positive density dependence at all densities, and another range where positive density-dependent mortality is very low even at very low densities (one might not consider those cases as conspecific positive density dependence). The dashed red lines correspond to the $s$ values above which increased density-dependence factor ($s$) can favor coexistence (see *Supplementary file 1B*). Note the use of a logarithmic y-scale; the dashed red lines are not planes of symmetry. Orange dots corresponds to the combinations of parameters tested in *Figure 5*. Densities of the least competitive species at coexistence equilibrium are also represented (**d–f**). The growth rate of the most competitive species is not affected by the other species. Therefore, at the coexistence equilibrium, the density of the most competitive species correspond to the density of the least competitive species if $\alpha_1 = 0$.

The online version of this article includes the following figure supplement(s) for figure 4:

**Figure supplement 1.** Densities at coexistence equilibrium.
**Figure supplement 2.** Identification of the coexistence equilibrium using analytical derivations or using the numerical resolution method.
**Figure supplement 3.** With other non-linear mortality functions.
**Figure supplement 4.** With a linear mortality function.

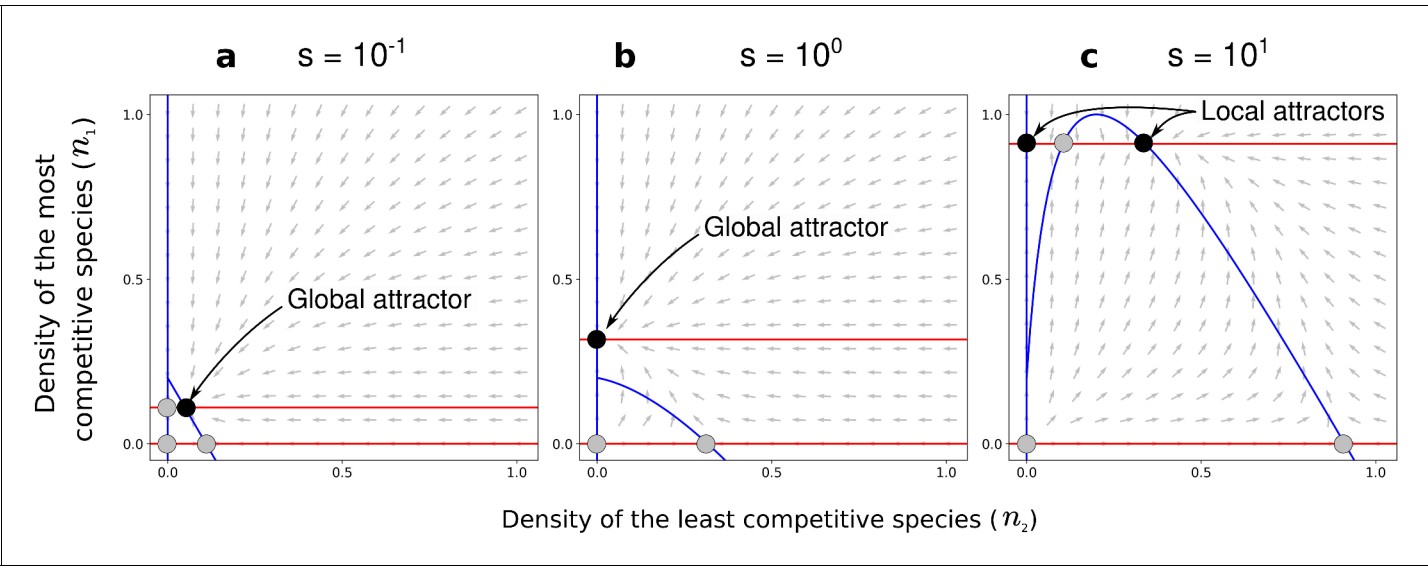

**Figure 5.** Effect of the positive density-dependence factor (*s*) on the zero-net-growth isoclines in the case of asymmetric competition for resources. Gray arrows represent the directions of the deterministic changes of species densities. Red and blue lines correspond to the isoclines (when $\mathrm{d}n_i/\mathrm{d}t = 0$) for species 1 and 2, respectively. Black and gray points represent stable and unstable equilibria, respectively. In each panel, the nature of the stable equilibria is annotated. Panel (**a**) corresponds to a case with a coexistence equilibrium that is a global attractor, panel (**b**) to a case without coexistence equilibrium, and panel (**c**) to a case with a coexistence equilibrium that is a local attractor. Other parameters: $d = 0.9$ and $\alpha_1 = 0.5$, corresponding to the combinations of parameters represented by orange dots in **Figure 4c** and **Figure 4f**.

without conspecific positive density dependence (when $s = 0$; **Holt, 1985**). In other words, the coexistence equilibrium fulfills the condition of global stability – i.e. the coexistence equilibrium is a *global attractor* (white zone in **Figure 4a–c**; e.g., **Figure 5a**). For a high density-dependence factor $s$, however, the least competitive species cannot invade. The coexistence equilibrium does not fulfill the condition of global stability – i.e. the coexistence equilibrium is a *local attractor* (gray zone in **Figure 4a–c**; e.g., **Figure 5c**). Therefore, increased density-dependence factor $s$ can favor the existence of a coexistence equilibrium that is not attained if the least competitive species is initially too rare – i.e. it can increase the feasibility of coexistence, despite inhibiting global stability.

At coexistence equilibrium, the least competitive species is always at lower density than the most competitive species (**Figure 4d–f**). Therefore, one might expect that asymmetric competition and density-dependent mortality should act synergistically and complementarily to promote competitive exclusion, thereby inhibiting the feasibility of coexistence. Yet, this model predicts a non-linear effect of the density-dependence factor (*s*) on the feasibility of coexistence. A feasible coexistence equilibrium arises for high density-dependence factors. The reason is simple: species at high density do not necessarily benefit the most from increased density-dependence factor (see pairs of arrows representing changes in mortality with increased $s$, in **Figure 3**). For example, in the case of extreme density-dependence factors, species at high density do not benefit from increased density-dependence factor because they do not incur density-dependent mortality (e.g. $D(1) \simeq 0$ for $s = 10^4$). Mathematically, this effect can be made clear by considering the non-linearity of the partial derivative $\frac{\partial D}{\partial s}$ as a function of $n_i$; increased density-dependence factor only slightly decreases the mortality rate of species at high density ($\frac{\partial D}{\partial s}$ close to 0 for high $n_i$, **Figure 3—figure supplement 2** and **Figure 3—figure supplement 3**). Now, suppose that coexistence occurs, with the least competitive species being at lower density than the most competitive species. In that case, by greatly decreasing the mortality rate of the least abundant species, increased density-dependence factor $s$ increases the density of the least competitive species (**Figure 4d–f** and **Figure 4—figure supplement 1**), and can therefore facilitate its maintenance. In conditions where the least competitive species is excluded, this same effect can promote the feasibility of coexistence.

This phenomenon is not specific to extreme cases of conspecific positive density dependence; increased density-dependence factor can increase the feasibility of coexistence among competing

species even for intermediate density-dependence factors (dashed red line at $s<10^2$ in *Figure 4*), where there is still a non-linear positive gain due to high density (i.e. saturation does not occur at very low density as for $s = 10^4$, see *Figure 3*). Notably, this phenomenon occurs when other non-linear or linear mortality functions are implemented (using a numerical method validated in *Figure 4—figure supplement 2*; *Figure 4—figure supplement 3* and *Figure 4—figure supplement 4*).

Overall, conspecific positive density dependence characterized by a high factor $s$ leads to coexistence that is a local attractor, which is often considered a 'weaker' form of coexistence, because it cannot be assembled easily through invasion, and because it is not robust to stochasticity and strong perturbation (*Chesson, 2000*). Additionally, conspecific positive density dependence characterized by an intermediate factor $s$ strongly inhibits the feasibility of coexistence (which is globally stable without positive density dependence, when $s$ is close to 0). In this model with asymmetric competition for resources, conspecific positive density dependence is therefore best seen as a mechanism inhibiting coexistence. Yet, as we shall see in the next section, conspecific positive density dependence associated with reduced mortality can substantially increase the feasibility domain of the coexistence equilibrium when competitive exclusion is driven by other forms of species asymmetry. This relies on the same effect of positive density dependence on coexistence than the one characterized analytically above.

## Two-species models with other forms of competitive exclusion
### Models
I now test whether conspecific positive density dependence acting on mortality increases the feasibility of coexistence in models accounting for other forms of species asymmetry. Each of those models include (1) symmetric competition for resources, (2) asymmetry among species, and (3) a positive density-dependent mortality term. Asymmetry among species takes the form of either differences in basal mortality rates (*Equation 2*) or asymmetric reproductive interference (*Equation 3*). All other assumptions are the same as in the model with asymmetric competition for resources analyzed above. These systems of equations include additional linear and nonlinear terms, making analytical resolutions difficult. The conditions of existence of the coexistence equilibrium as a local or global attractor are therefore assessed using a numerical method (full methods appear in *Supplementary file 1C*). This method is validated by applying it to the model described in the previous section and by comparing the results to the analytical ones (*Figure 4—figure supplement 2*).

In the model with differences in basal mortality rates, population dynamics are governed by the equations:

$$\begin{cases} \dfrac{dn_1}{dt} = n_1[1 - n_1 - \alpha n_2 - d \times D(n_1)] \\ \dfrac{dn_2}{dt} = n_2[1 - n_2 - \alpha n_1 - [d + \delta(1-d)] \times D(n_2)] \end{cases} \tag{2}$$

Parameter $\alpha \in [0,1]$ corresponds to the intensity of symmetric competition for resources (heterospecific negative density dependence). Parameter $\delta \in [0,1]$ represents the difference in basal mortality rate between species. If $\delta = 0$, the two species have the same basal mortality rate $d$. If $\delta > 0$, species 2 incurs a higher basal mortality rate than species 1. Therefore, $\delta$ reflects the intensity of species asymmetry, just like parameter $\alpha_1$ in *Equation 1a*. The positive density-dependent mortality function $D(n_i)$ is modeled as in *Equation 1b* and is characterized by its density-dependence factor $s$.

In the model with asymmetric reproductive interference, population dynamics are governed by the equations:

$$\begin{cases} \dfrac{dn_1}{dt} = n_1[1 - n_1 - \alpha n_2 - d \times D(n_1)] \\ \dfrac{dn_2}{dt} = n_2\left[\dfrac{1}{1+\alpha_1'\frac{n_1}{n_2}} - n_2 - \alpha n_1 - d \times D(n_2)\right] \end{cases} \tag{3}$$

Here, parameter $\alpha_1' \in [0,1]$ reflects the intensity of asymmetric reproductive interference (*Yoshimura and Clark, 1994*; *Kishi and Nakazawa, 2013*), assuming that sexual interactions are stronger among conspecifics than among heterospecifics, hence $\alpha_1' \leq 1$. If $\alpha_1' = 0$, there is no reproductive interference. If $\alpha_1' > 0$, interspecific sexual interactions reduce the reproductive success of

females of species 2 – i.e. the presence of species 1 inhibits the growth rate of species 2 such that $\frac{1}{1+\alpha'_1\frac{n_1}{n_2}}<1$. Contrary to competition for resources that is density-dependent, reproductive interference is frequency-dependent (*heterospecific negative frequency dependence*); the presence of few heterospecifics might substantially decrease reproductive success as long as they are more abundant than conspecifics, ultimately leading to species exclusion (hence the term $\frac{n_1}{n_2}$ in the denominator, following *Kishi and Nakazawa, 2013*).

## Numerical resolutions

Depending on the parameters considered, those systems can be characterized by a single stable coexistence equilibrium (white and gray zones in *Figure 6*). Depending on the form of species asymmetry, the feasibility domain of species coexistence is more or less constrained. In particular, reproductive interference is more prone to species exclusion than the other forms of species asymmetry tested (e.g. see comparable graphs in *Figure 6—figure supplement 1* vs. *Figure 6—figure supplement 2*). Likewise, under asymmetric reproductive interference which is frequency-dependent, coexistence is only ever a local attractor; the least competitive species cannot invade if it is too rare initially and the condition of global stability is never fulfilled (as shown by *Kishi and Nakazawa, 2013*). Importantly, in those two models, the least competitive species can be excluded when there is no positive density dependence (when $s$ is close to 0, *Figure 6*), contrary to the case with asymmetric competition for resources. This is not surprising: compared to *Equation 1a*, the expression of $\mathrm{d}n_2/\mathrm{d}t$ is characterized by an additional negative term ($-n_2 \times \delta\,(1-d) \times D(n_2)$) in *Equation 2* or by a low intrinsic growth rate ($n_2 \times \frac{1}{1+\alpha'_1\frac{n_1}{n_2}} \leq n_2$) in *Equation 3*. Those additional terms governing population dynamics favor the exclusion of the least competitive species, despite the reduced growth rate of the most competitive species (term $n_1 \times (-\alpha\,n_2)$ in *Equations 2 and 3*.

For asymmetric basal mortality and asymmetric reproductive interference, the density-dependence factor ($s$) has a similar effect on species coexistence as under asymmetric competition for resources. Conspecific positive density dependence characterized by an intermediate factor $s$ can lead to the loss of the coexistence equilibrium (black zone in *Figure 6a–c*); in particular, this occurs under asymmetric reproductive interference if there is some symmetric competition for resources (for $\alpha = 0.5$ in *Figure 6—figure supplement 2*). More importantly, under the two forms of species asymmetry implemented here, conspecific positive density dependence characterized by a high factor $s$ favors the existence of the coexistence equilibrium as a local attractor (*Figure 6*). Indeed, conspecific positive density dependence characterized by a high factor $s$ reduces the mortality of the least competitive species (as shown in the case of asymmetric competition for resources; see *Figure 4d–f*) and thereby increases the feasibility of coexistence. This effect is made clear in supplementary analyses where mortality is not density-dependent and where increased $s$ associates with reduced mortality (top rows in *Figure 6—figure supplement 3* and *Figure 6—figure supplement 4*).

In the previous model with asymmetric competition for resources, the coexistence equilibrium is either lost or non-globally stable under conspecific positive density dependence associated with reduced mortality (compared to the coexistence equilibrium when $s$ is close to 0, *Figure 4*). Here, this form of positive density dependence can substantially increase the coexistence region, compared to the situation without positive density dependence (*Figure 6*). Note, however, that the condition of global stability is never fulfilled when positive density dependence favors the feasibility of coexistence.

In supplementary analyses, I implemented other values of parameter $\alpha$ (*Figure 6—figure supplement 1* and *Figure 6—figure supplement 2*) or other non-linear or linear mortality functions (*Figure 6—figure supplement 3*, *Figure 6—figure supplement 4*; *Figure 6—figure supplement 5*, and *Figure 6—figure supplement 6*). Just like in the main analyses, conspecific positive density dependence associated with reduced mortality can substantially increase the coexistence region.

## Multispecies models
### Models

To finish, I assess the effects of conspecific positive density dependence on the maintenance of species-rich communities in multispecies models. The previous two-species models are combined and

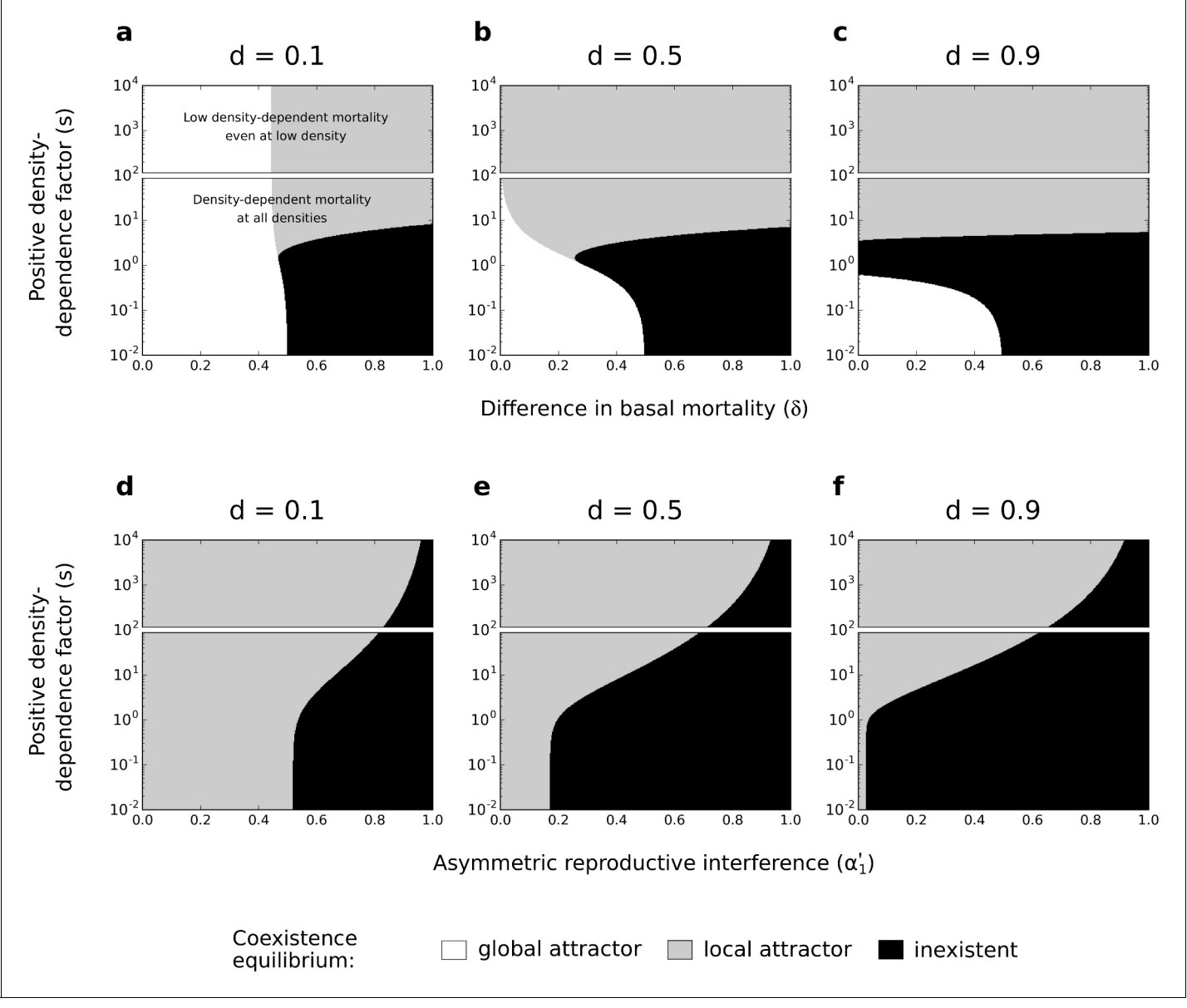

**Figure 6.** Effects of difference in basal mortality between species ($\delta$) (**a,b,c**) or asymmetric reproductive interference ($\alpha_1'$) (**d,e,f**), and positive density-dependence factor ($s$) on coexistence, using the numerical resolution method (systems of *Equations 2 and 3*, respectively). If the coexistence equilibrium exists, it is either a global attractor or a local attractor. In each subfigure, the range of $s$ is arbitrarily divided in two; one range where there is conspecific positive density dependence at all densities, and another range where positive density-dependent mortality is very low even at very low densities. Because those two forms of species asymmetry do not promote species exclusion to the same extent, the levels of symmetric competition for resources are chosen as: $\alpha = 0.5$ (**a,b,c**) or $\alpha = 0$ (**d,e,f**).

The online version of this article includes the following figure supplement(s) for figure 6:

**Figure supplement 1.** Difference in basal mortality with other levels of symmetric competition for resources.
**Figure supplement 2.** Asymmetric reproductive interference with other levels of symmetric competition for resources.
**Figure supplement 3.** Difference in basal mortality with other non-linear mortality functions.
**Figure supplement 4.** Asymmetric reproductive interference with other non-linear mortality functions.
**Figure supplement 5.** Difference in basal mortality with a linear mortality function.
**Figure supplement 6.** Asymmetric reproductive interference with a linear mortality function.

modified to account for an arbitrary number of species. The population dynamic of each of the $N$ species is:

$$\frac{dn_i}{dt} = n_i \left[ \frac{1}{\sum_{j=1}^{N} \alpha'_{j,i} \frac{n_j}{n_i}} - \sum_{j=1}^{N} \alpha_{j,i} n_j - [d + \delta_i (1-d)] \times D(n_i) \right] \tag{4}$$

Parameters $\alpha_{j,i}$ and $\alpha'_{j,i}$ represent the strengths of resource competition and reproductive interference experienced by species $i$ because of the presence of species $j$ ($\forall i$, $\alpha_{i,i} = 1$ and $\alpha'_{i,i} = 1$). Parameter $\delta_i$ represents the increase in mortality rate of species $i$ compared to the minimum basal mortality rate $d$. The positive density-dependent mortality function $D(n_i)$ is modeled as in *Equation 1b*, and is characterized by its density-dependence factor $s$.

Four different scenarios are considered successively: (1) only asymmetric competition, (2) only asymmetric reproductive interference, (3) only differences in basal mortality, (4) random communities with all forms of asymmetry. With the first three scenarios, I assess the robustness of the predictions of the two-species models analyzed above, and with the fourth scenario, I test whether conspecific positive density dependence acting on additional mortality can help maintain a high species richness in a random community. In scenarios 2 and 3, I assume that there is symmetric competition for resources ($\alpha_{i,j} = \alpha$ if $i \neq j$), just like in the corresponding two-species models (full methods appear in *Supplementary file 1D*).

## Numerical simulations

For each scenario, 500 species pools of 100 species each were constructed by drawing species parameter values from arbitrary uniform distributions (detailed in *Supplementary file 1D*; see the robustness to variations in the uniform distributions tested in *Figure 7—figure supplement 1*). All species were equally abundant initially ($n_i(0) = 1$), and simulations were run long enough for initial transients to dissipate. Species were declared extinct if their density fell below $10^{-3}$. At the end of each simulation, the number of species remaining was recorded. In all simulations in which multiple species persisted, coexistence occurred at a stable equilibrium.

Consistent with the previous two-species models, the density-dependence factor has a non-linear effect on species richness maintained at equilibrium (*Figure 7a–c*). Indeed, species richness remains high under the conditions where there is a coexistence equilibrium in the two-species models, including for high density-dependence factors (*Figure 4* and *Figure 6*). Finally, in simulations with random communities of species with all forms of asymmetry, conspecific positive density dependence characterized by a high factor $s$ can favor the existence of stable equilibrium points characterized by the maintenance of many competing species (*Figure 7d*). Therefore, conspecific positive density dependence acting on extrinsic mortality can help maintain species-rich communities.

In supplementary analyses, I implemented other values of parameters $d$ and $\alpha$ (*Figure 7—figure supplement 2*, *Figure 7—figure supplement 3*, *Figure 7—figure supplement 4* and *Figure 7—figure supplement 5*), other non-linear or linear mortality functions (*Figure 7—figure supplement 6* and *Figure 7—figure supplement 7*), or just a fraction of species undergoing conspecific positive density dependence (*Figure 7—figure supplement 8*). Just like in the main analyses, conspecific positive density dependence acting on extrinsic mortality can help maintain species-rich communities. This is also the case when species parameter values are drawn from other uniform distributions (*Figure 7—figure supplement 1*).

A second set of simulations in which species were introduced one at a time produced different results. In such scenarios of community assembly, positive density dependence does not help produce species-rich communities (if species are introduced at very low density; *Figure 7—figure supplement 9*). This is consistent with the two-species models analyzed where the condition of global stability is not fulfilled for a high density-dependence factor.

## Discussion

When intraspecific competition is stronger than interspecific competition, resulting negative density-dependent growth rate has long been recognized as an important factor favoring coexistence among competing species (*MacArthur, 1970*; *Chesson, 2000*; *McPeek, 2012*), and it is therefore not surprising that conspecific positive density dependence acting on intrinsic growth rate can inhibit

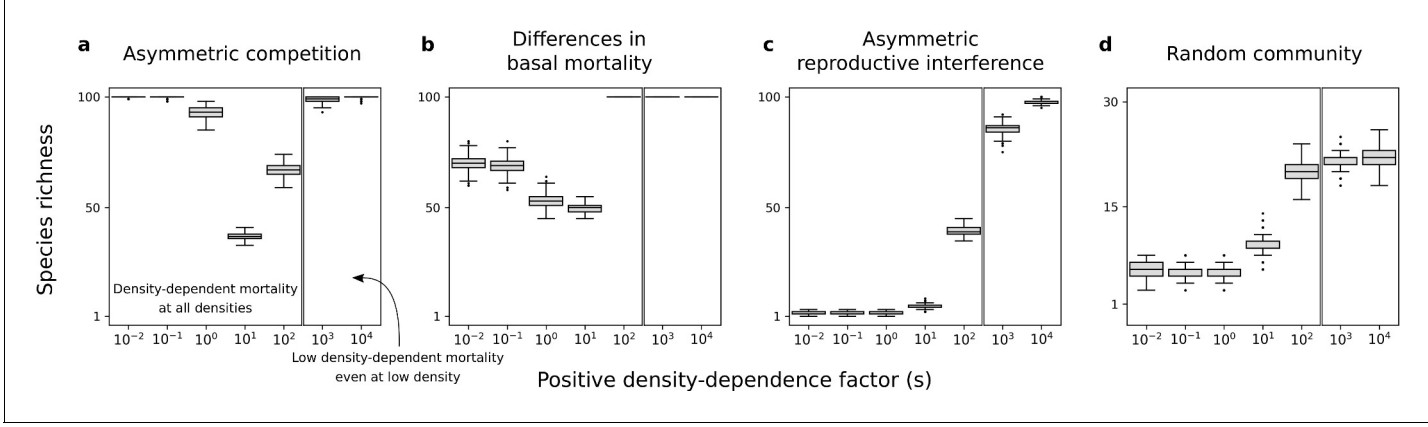

**Figure 7.** Species richness maintained in simulated communities. Each box represents the distribution for 500 simulated communities with 100 species initially. (**a**) Only asymmetric competition for resources. (**b**) Only differences in basal mortality. (**c**) Only asymmetric reproductive interference. (**d**) Random communities with all forms of species asymmetry. In each subfigure, the range of $s$ is arbitrarily divided in two; one range where there is conspecific positive density dependence at all densities, and another range where positive density-dependent mortality is very low even at very low densities. Minimum basal mortality rate: $d = 0.5$; Levels of symmetric competition for resources: $\alpha = 0.05$ (**b**) or $\alpha = 0$ (**c**), competition for resources is asymmetric otherwise (**a, d**).

The online version of this article includes the following figure supplement(s) for figure 7:

**Figure supplement 1.** With parameters drawn from other uniform distributions.
**Figure supplement 2.** Asymmetric competition for resources with other minimum basal mortality rates.
**Figure supplement 3.** Differences in basal mortality with other minimum basal mortality rates and other levels of symmetric competition for resources.
**Figure supplement 4.** Asymmetric reproductive interference with other minimum basal mortality rates and other levels of symmetric competition for resources.
**Figure supplement 5.** Random communities with other minimum basal mortality rates.
**Figure supplement 6.** Random communities with other non-linear mortality functions.
**Figure supplement 7.** Random communities with a linear mortality function.
**Figure supplement 8.** Random communities with only a fraction of species incuring conspecific positive density dependence.
**Figure supplement 9.** Random communities when species are introduced one at a time.

coexistence among competing species (as shown theoretically by *Wang et al., 1999*; *De Silva and Jang, 2015*). Nonetheless, a variety of mechanisms can generate positive density-dependent growth rates (*Courchamp et al., 1999*; *Stephens et al., 1999*; *Kramer et al., 2009*). Contrary to expectations, the models analyzed in this paper suggest that conspecific positive density dependence associated with reduced mortality (e.g. reduced predation via cooperative defense, predator satiation or aposematism; *Figure 2*) can favor the maintenance of species-rich communities.

How can different forms of conspecific positive density dependence have opposite effects on the feasibility of species coexistence? Conspecific positive density dependence *per se* promotes species exclusion by reducing the growth rate of inferior competitors that are at lower density than superior competitors. This inhibits species coexistence in previous models accounting for conspecific positive density dependence (*Wang et al., 1999*; *De Silva and Jang, 2015*), but also in the models analyzed in this paper (for intermediate factor $s$). However, reduced mortality associated with conspecific positive density dependence can also increase the feasibility domain of species coexistence (i.e. it can favor the existence of a locally stable equilibrium of coexistence). This outcome relies on the effect of increased mortality on population dynamics. Increased mortality can lead to the exclusion of less competitive species (*Abrams, 1977*; *Holt, 1985*; but see *Abrams, 2001*), and this is the case in the two models analyzed numerically in this paper (*Equations 2 and 3*). Therefore, by reducing the mortality of less competitive species, conspecific positive density dependence can favor coexistence among competing species. By contrast, conspecific positive density dependence associated with an increase of mortality should favor species exclusion (e.g. anthropogenic Allee effects; *Courchamp et al., 2006*). Likewise, in cases where increased mortality increases the feasibility of coexistence (e.g. in a MacArthur's consumer-resource model; *Abrams, 2001*), conspecific positive density dependence associated with reduced mortality should also favor species exclusion.

Under conspecific positive density dependence, the least competitive species is often not able to increase in density if it is too rare – i.e. coexistence is only a *local attractor* and the condition of global stability is not fulfilled. Invasibility is often considered as a fundamental criterion for species coexistence regardless of the underlying mechanism (*Chesson, 2000*; *Siepielski and McPeek, 2010*; *Grainger et al., 2019*); indeed, global stability of a feasible equilibrium point is a sufficient condition for species coexistence. Nonetheless, the condition of global stability is rarely fulfilled in systems with more than two species, and the feasibility of coexistence has also been recognized as an important determinant of multispecies coexistence (see *Saavedra et al., 2017*; *Levine et al., 2017*; *Grainger et al., 2019*, for discussion on the evaluation of multispecies coexistence). While the coexistence equilibria identified do not satisfy this invasibility criterion, conspecific positive density dependence strongly increases the coexistence region (i.e. the feasibility domain of coexistence) in the two-species models analyzed numerically. Moreover, numerical simulations of multispecies models show that conspecific positive density dependence can favor the existence of stable equilibrium points characterized by the maintenance of many competing species. This suggests that positive density dependence acting on mortality can help maintain species diversity in ecological communities.

One aspect that has not been investigated in this study is the implication of stochasticity for the maintenance of species-rich communities (all numerical simulations were deterministic). In the models analyzed in this study, conspecific positive density dependence can only increase the feasibility of a local coexistence equilibrium point that is not robust to stochasticity and strong perturbations. However, another effect of conspecific positive dependence is to increase the species densities at coexistence equilibrium (see *Figure 4* and *Supplementary file 1A*), and therefore to increase robustness to stochastic changes. Under conspecific positive density dependence, only strong stochasticity may lead to local extinction, yet such strong stochasticity would also favor the invasion of new species – the threshold above which a species can invade would be more easily attained by chance. In the face of stochasticity, the effect of conspecific positive density dependence on species richness in metacommunities may therefore differ from the deterministic case studied in this paper, and therefore requires further theoretical investigations (e.g. following the approach followed by *Schreiber et al., 2019*, combining frequency-dependence and environmental stochasticity).

Putative examples of conspecific positive density dependence acting on extrinsic mortality have already been described in many taxa (*Figure 1* and *Figure 2*) (note however that difficulties in detecting negative density dependence may be similarly applied in detecting positive density-dependence; *Detto et al., 2019*). This form of conspecific positive density dependence may play an important role in structuring certain ecological communities. For instance, the high species richness that can be found in aposematic organisms at local scale (e.g. in aposematic butterflies, *Willmott et al., 2017*) is likely to be caused by conspecific positive density dependence associating with reduced mortality (via density-dependent predator avoidance which has received much empirical support; *Ruxton et al., 2018*). By contrast, an overall pattern of negative density dependence has been observed in other ecological communities (e.g. in many plant communities; *Adler et al., 2018*). This does not mean that conspecific positive density dependence associated with reduced mortality is not pervasive in those communities. Indeed, it is important to remember that different mechanisms leading to density dependence can occur simultaneously in the same population (*Berec et al., 2007*). In particular, density-dependent mechanisms can counteract with each other (*Feldman and Morris, 2011*; *Bergamo et al., 2020*), and observing an overall pattern of negative density dependence does not mean that some forms of positive density dependence do not take place. In the models analyzed in this paper, conspecific positive density dependence associated with reduced mortality promote species-rich communities in concert with negative density dependence (driven by intraspecific competition for resources), highlighting the necessity of dissecting the different density-dependent mechanisms acting simultaneously (as in *Feldman and Morris, 2011*; *Bergamo et al., 2020*). In particular, experimental manipulations of population density coupled with other treatments (e.g. removal of predators/herbivores) could prove useful not only to infer the nature and strength of density dependence but also to define the underlying mechanisms (e.g. this is how negative density dependence in populations of reef fishes has been shown to be driven by competition and predation; *Hixon, 1997*; *Forrester and Steele, 2000*; *Carr et al., 2002*; *Holbrook and Schmitt, 2002*).

In addition to the assessment of conspecific positive density dependence associated with reduced mortality in nature, the causal link between this form of density dependence and species coexistence remains to be tested empirically. Imposing and amplifying conspecific positive density dependence in a controlled experimental setup could prove fruitful (as in *Williams and Levine, 2018*, where negative density dependence was manipulated). For instance, within an experimental setup with two competing species, imposing additional mortality (by removing individuals artificially) and enforcing positive density dependence on this source of mortality (with different decline rates of mortality with density; as in *Figure 3*) could provide information on the implication of conspecific positive density dependence associated with reduced mortality for coexistence. In the theoretical models analyzed here, conspecific positive density dependence strongly promoted the maintenance of species undergoing asymmetric reproductive interference. I thus encourage testing the theoretical predictions presented in this paper on empirical data in organisms involved in such sexual interactions (e.g. *Takafuji et al., 1997*; *Kishi et al., 2009*; *Takakura et al., 2009*; *Crowder et al., 2010*).

## Conclusion

Although a variety of coexistence mechanisms operate simultaneously in nature, these modeling results suggest that conspecific positive density dependence play a considerable role in structuring ecological communities. Conspecific positive density dependence may have opposite effects on species coexistence depending on the underlying density-dependent mechanism. In particular, contrary to expectations, conspecific positive density dependence associated with reduced mortality can help maintain species-rich communities. Hopefully, this theoretical analysis will stimulate further empirical research precisely testing the prevalence of conspecific positive density dependence and investigating its implications for population dynamics, including for coexistence among competing species.

## Acknowledgements

I am very grateful to M Dubard, T Koffel and P Quévreux for comments on early versions of the manuscript. I also thank David Donoso and three anonymous reviewers for comments that have helped improve this manuscript.

## Additional information

### Funding

No external funding was received for this work.

### Author contributions

Thomas G Aubier, Conceptualization, Software, Formal analysis, Validation, Investigation, Visualization, Methodology, Writing - original draft, Writing - review and editing

### Author ORCIDs

Thomas G Aubier (iD) https://orcid.org/0000-0001-8543-5596

### Decision letter and Author response

Decision letter https://doi.org/10.7554/eLife.57788.sa1
Author response https://doi.org/10.7554/eLife.57788.sa2

## Additional files

### Supplementary files

- Supplementary file 1. Analytical derivations and supplementary information.
- Source code 1. Simulation code.
- Transparent reporting form

## Data availability

All data generated or analysed during this study are included in the manuscript and supporting files. The code (Python) is made available as Source code 1.

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
