## [Decision Letter]

**Acceptance summary:**

For decades, negative density dependence (NDD) has been a major paradigm guiding theoretical and experimental research on mechanisms maintaining biodiversity. This paper goes outside the box and provides mathematical support for positive density dependence (PDD) as an additional viable force maintaining biodiversity. By reducing mortality, PDD could limit the exclusion of less competitive species by dominant ones. We are sure that this manuscript will inspire both theoreticians and field biologists aiming to understand how species to coexist in nature.

**Decision letter after peer review:**

[Editors’ note: the authors submitted for reconsideration following the decision after peer review. What follows is the decision letter after the first round of review.]

Thank you for submitting your work entitled "Positive density-dependence acting on mortality can help maintain species-rich communities" for consideration by *eLife*. Your article has been reviewed by three peer reviewers, and the evaluation has been overseen by a Reviewing Editor and a Senior Editor. The reviewers have opted to remain anonymous.

Our decision has been reached after consultation between the reviewers. Based on these discussions and the individual reviews below, we regret to inform you that your work will not be considered further for publication in *eLife*. We all agree, however, that an improved version of the manuscript (addressing completely reviewer's points below) may be submitted as a new submission in *eLife*.

All reviewers find merit in your study. And all of them suggest that more theoretical work (like yours) is needed to expand the field, given that positive density-dependence is so often observed in nature. The manuscript is inspiring. For example, one of the reviewers highlighted: "This manuscript is very creative and thought-provoking because it makes me think more about very steep positive density-dependent gradients in a new light, as facilitating coexistence rather than being a unique part of species' ecology". The reviewers also applaud the author's efforts in proposing potentially new mechanisms of species coexistence in light of positive density-dependence. However, the author needs to make more efforts in building biological motivation and checking the robustness of the results. The manuscript is too theoretical/mathematical for the *eLife* audience and lacks more-biological 'real-life' examples.

Reviewer #1:

I think Aubier accomplished an interesting modeling manuscript on an interesting topic, positive density dependence acting on mortality. However, I feel the manuscript is too theoretical/mathematical for *eLife* and lacks a more-biological 'real-life' examples.

I do think that a glossary, and a comparative table with NDD on the left, and PDD on the right, then specific patters (like PDD on mortality) on a second row, then possible mechanisms responsible for these patterns, outcomes, and finally examples in real life, could be something to add.

I think the author can do more to have an easily understandable manuscript. All, the mathematical formulas, the graphs, and the text itself are very difficult to follow. Even the code in the zip file is not annotated. This does not have to be this way. Also, some re-ordering may be also appropriate (the first paragraph in the Discussion should likely be the first of the Introduction), and many ideas are repeated two or more times (i.e., Positive density-dependence has been described for most major animal taxa…). Some combinations of words, like *existence of coexistence*, just make your manuscript unnecessarily difficult to follow. And discussion on mimicry seems just speculation at this point.

Sorry for self-advertizing, but in one of my papers, we forced NDD to Devil's Gardens, one case where ants decrease plant mortality. So patches of monospecific plants accumulate in the forest. We really had a hard time explaining this non-NDD case, (I guess this is then a case of PDD). What I want to say is that there are potentially hundreds of ecologists out there dealing with similar issues in their own system. This manuscript could benefit them. Currently, it does not.

Reviewer #2:

This manuscript is very creative and thought-provoking because it makes me think more about very steep positive density-dependent gradients in a new light, as facilitating coexistence rather than being a unique part of species' ecology. For example, bighorn sheep suffer strong Allee effects because small groups are depredated quickly by mountain lions. This is a terrific example of the Allee effect, but one rarely thinks of this sort of positive density dependence as being a mechanism of species coexistence between large ungulates. This paper makes me think about that.

I do think that the model needs more contextualization in biology from top to bottom. I had to read very carefully, and twice, to understand what I did – for *eLife*, the paper has to sing so that readers get most of it on the first read. There is an opportunity, and I would argue a necessity, to clarify intra- and interspecific density-dependent processes in the Introduction. The choice of terms in the modeling equations themselves could also be better tied to biology to explain exactly what kinds of systems they represent, and how broadly they can be interpreted. My specific comments are below:

1) Negative density dependence favors coexistence if intraspecific density dependence is stronger than interspecific density dependence. Positive density-dependence seems like a strange mechanism to promote species coexistence when it is interspecific because the most abundant species should only become a stronger and stronger competitor as it rises in abundance. Intraspecific positive density dependence – e.g. Allee effects – seems to make it more difficult for rare species to invade, so it does not promote global stability (subsection “Analytical resolution”, second paragraph). As I understand the analytical part of the model (and I have to be honest, I only read Supplementary file 1A), positive density dependence is only intraspecific because the D(ni) function for each species only includes individuals of that species. s it is always the same between species, though so their density dependence varies perfectly in tandem. I think this is really what struck me as counterintuitive because if I were to think about positive density dependence in the context of competition, I would imagine it to operate interspecifically as well so that the more abundant a species was, the better it would be able to outcompete its competitors (45-47). If I am right about what the model means, then it might be very helpful in the Introduction to explain not just "positive density dependence" writ large, but the different ways it might function (e.g. interspecifically versus intraspecifically, or using specific natural history examples that the model applies to and contrasting them to natural history examples where it does not). The fourth paragraph of the Introduction starts to do this, but are very abstract and don't go into enough detail about differences in kinds of positive density dependence, only its general effects.

2) At very high values of positive frequency dependence, the effects of positive density dependence essentially disappear because the populations hit zero mortality so quickly. I could imagine this being very important in cases where there are for example two aposematic species and both have sufficiently high population sizes that both are protected from predation. Perhaps it would also make sense to imagine two species of pack-hunting carnivores that have zero success alone, but much greater success with even a few individuals, so the two species might be able to coexist as long as both are already present in sufficient abundance. This is sort of an Allee effect too, of course, which are also experienced by species of herd animals that cannot survive in small groups, because they are too vulnerable to predation. But this model seems to describe species that are very similar in their ecology, not just because they are forced to be so to analyze the effects of density dependence, but also because density has identical intraspecific effects. Is this true? If so it would be helpful to explain in the text, and if not, then it would be helpful to explain how broadly it applies.

3) In (2), has the carrying capacity – or effective carrying capacity – also changed? Is the purpose of this model to contrast with (1), where one spp. is a superior competitor, whereas here in (2), one spp. is a superior reproducer/survivor?

4) For Equation 3, I need more hand-holding to explain what's going on. It looks like the new term n2/(n2 + a'1*n1) means the frequency of individuals in the community that is composed of n2 – is that correct? So is this a way of incorporating positive interspecific frequency dependence into your model that already includes positive intraspecific density dependence? Would there be any difference if you modeled reproductive interference as an effect on d rather than on ni, since in a literal way it should affect reproduction rather than carrying capacity? I could also imagine it not matter if its influence is only supposed to be manifest by decreasing dn2/dt, though.

Reviewer #3:

Positive density-dependence is widely observed in nature. It is widely believed that positive density-dependence inhibits species coexistence. The main result of this manuscript is that a positive density-dependence on mortality, under certain forms of population dynamics, increases the feasibility domain of species coexistence. The paper is well-written (but I am not sure if the math here would be easily accessible to the majority of the readership at *eLife*). I applaud the author's efforts in proposing potentially new mechanisms of species coexistence by positive density-dependence, which could be an important contribution to the coexistence theory. However, the author needs to make more efforts in building biological motivation and checking the robustness of the results.

1) The biological relevance.

– The author motivated the paper by pointing up the ubiquitousness of positive density-dependence (Introduction, fifth paragraph). However, if I understood correctly, most references focus on the Allee effect of the birth rates. Then, is there substantive literature of empirical evidence that positive density-dependence acts on mortality? To be clear, I am not saying that there is not, but to encourage the author to be more explicit about the biological motivations.

– In the fifth paragraph of the Discussion, the author concluded that it is yet impossible to validate the results. I apologize if I misunderstood, but I feel that the paper offers no guidelines for empirical tests. To be clear, I am not asking the author to run analysis on some empirical data, but to encourage the author to discuss how this theory can be potentially tested. Otherwise, it may leave the reader the impression of a math exercise disconnected from ecological nature.

2) The robustness of the results. I understand the following requests can be a bit much, but I sincerely believe that when one proposes a new ecological mechanism, the author needs to prove that it is theoretically robust to convince the empiricists. However, I am happy for a discussion if the author disagrees with anything follows.

– This paper only studies one particular functional form of mortality with positive density-dependence. As the author stated that "assessing the functional form of positive density-dependence in natural populations is tedious", I think the author needs to study other functional forms of positive density-dependence, at least in the 2-species case.

– This paper does not have a sort of null model to estimate the effects of positive density-dependence. That is, whether the effects of positive density-dependence solely comes with the non-linear functional form or because of the positiveness. To do this, I suggest the author add some additional tests on negative density-dependence with the same functional response, to see that the effects are actually coming from positiveness.

– This paper considers the same functional form of positive density-dependence for every species. However, I think it is worth examining some with positive density-dependence and some without in multi-species case.

– To be honest, I am a bit lost why section 1 exists. I thought the author tries to prove that positive density-dependence helps coexistence, but why spent so much time on a particular case where positive density-dependence inhibits coexistence?

– I don't find the parametrization in Supplementary file 1D satisfactory. The range of parameters seems very arbitrary to me. I was puzzled that why no other parametrizations were examined.

---

## [Author Response]

[Editors’ note: the authors resubmitted a revised version of the paper for consideration. What follows is the authors’ response to the first round of review.]

All reviewers find merit in your study. And all of them suggest that more theoretical work (like yours) is needed to expand the field, given that positive density-dependence is so often observed in nature. The manuscript is inspiring. For example, one of the reviewers highlighted: "This manuscript is very creative and thought-provoking because it makes me think more about very steep positive density-dependent gradients in a new light, as facilitating coexistence rather than being a unique part of species' ecology". The reviewers also applaud the author's efforts in proposing potentially new mechanisms of species coexistence in light of positive density-dependence. However, the author needs to make more efforts in building biological motivation and checking the robustness of the results. The manuscript is too theoretical/mathematical for the eLife audience and lacks more-biological 'real-life' examples.

I thank the Editors for handling my manuscript and for their positive feedback. I am pleased that all reviewers approved my general aim of investigating the implications of conspecific positive density dependence for species coexistence, and appreciated the counter intuitiveness of my theoretical results. However, it is clear from the feedback I received that I had much more work to do to build biological motivation and to make my approach more impactful. In the revised version of this paper I have sought to address these concerns (i) by thoroughly defining the different types of density dependence (conspecific/heterospecific, negative/positive) that come into play in nature, (ii) by using natural history examples to illustrate the ubiquitousness of conspecific positive density dependence associated with reduced mortality, and (iii) by offering guidelines for testing empirically my theoretical predictions. Collectively, I hope these changes would make my study more accessible to both empiricists and theoreticians.

Additionally, I followed the reviewers’ suggestions of conducting further supplementary analyses (with more than 10 new supplementary figures); altogether, these analyses show that my results are robust to changes in the underlying modeling assumptions.

Reviewer #1:I think Aubier accomplished an interesting modeling manuscript on an interesting topic, positive density dependence acting on mortality. However, I feel the manuscript is too theoretical/mathematical for eLife and lacks a more-biological 'real-life' examples.

I thank reviewer #1 for this positive feedback. I have followed his/her suggestions, and I have made a considerable effort to make this theoretical study more easily accessible and therefore more impactful. By doing so, I hope that I addressed the reviewer’s concerns successfully.

I do think that a glossary, and a comparative table with NDD on the left, and PDD on the right, then specific patters (like PDD on mortality) on a second row, then possible mechanisms responsible for these patterns, outcomes, and finally examples in real life, could be something to add.

I followed the reviewer’s suggestion to add a Glossary defining terms describing species interactions, density dependence and frequency dependence (negative vs. positive, conspecific vs. heterospecific) (Box 1). In this new glossary but also throughout the manuscript, I am now very clear about the nature of each density-dependent process, and about how species interactions can lead to heterospecific density dependence. I also emphasize that I am investigating cases of conspecific positive density dependence (and not cases of heterospecific positive density dependence caused by mutualistic interspecific interactions) (Box 1).

Following the reviewer’s suggestion, I also added a comparative table (Figure 1). I decided to compare specific patterns of conspecific positive density dependence (associated with increased reproduction or reduced mortality) to emphasize the biological motivation behind my theoretical study on conspecific positive density dependence (note however that negative density dependence is described in Box 1). In this comparative table, I use natural history examples to illustrate each specific patterns (the references can be found in the excellent review papers cited in the caption of Figure 1). Additionally, I added some illustrations of specific natural history examples where conspecific positive density dependence associates with reduced mortality in different taxa (Figure 2) (Introduction).

Altogether, these amendments significantly clarify the manuscript and build the biological motivation behind this theoretical study.

I think the author can do more to have an easily understandable manuscript. All, the mathematical formulas, the graphs, and the text itself are very difficult to follow. Even the code in the zip file is not annotated. This does not have to be this way. Also, some re-ordering may be also appropriate (the first paragraph in the Discussion should likely be the first of the Introduction), and many ideas are repeated two or more times (i.e., Positive density-dependence has been described for most major animal taxa…). Some combinations of words, like existence of coexistence, just make your manuscript unnecessarily difficult to follow. And discussion on mimicry seems just speculation at this point.

In the revised version of the manuscript, I aimed at avoiding useless repetitions and unwieldy combinations of words (e.g., “existence of coexistence” is now referred to as “existence of equilibrium points characterized by species coexistence”). Likewise, I followed the suggestions of moving the paragraph describing the mechanisms favouring species coexistence from the Discussion to the Introduction.

I also removed the discussion on the maintenance of multiple mimicry rings, which was somehow off-topic.

Following the suggestions of the other reviewers, I clarified the presentation of the mathematical formulas by precisely using the terms describing density- or frequency-dependence (also defined in the Glossary; Box 1). In particular, I now precisely explain why I model reproductive interference as a frequency-dependent process acting on the growth rate (subsection “Two-species models with other forms of competitive exclusion”).

I apologize for the non-annotated code that I provided during the initial submission; the code is now fully annotated.

I hope that the modifications I made to address the other reviewers’ concerns also make the manuscript more understandable. Of course, I am open to any other suggestions that would make my study easier to follow.

Sorry for self-advertizing, but in one of my papers, we forced NDD to Devil's Gardens, one case where ants decrease plant mortality. So patches of monospecific plants accumulate in the forest. We really had a hard time explaining this non-NDD case, (I guess this is then a case of PDD). What I want to say is that there are potentially hundreds of ecologists out there dealing with similar issues in their own system. This manuscript could benefit them. Currently, it does not.

In the revised manuscript, I extended the Discussion to fill this gap. First, I advocate the need to assess the occurrence of conspecific positive density dependence in the wild. In particular, if empiricists observe an overall pattern of conspecific negative density dependence (as in many plant communities), some other forms of positive density dependence may still take place at the same time (and may promote species coexistence by their own). Second, I offer guidelines for testing empirically my theoretical predictions on the link between conspecific positive density dependence and coexistence. Altogether, I hope these amendments will be helpful to empiricists interested in the theoretical prediction presented in my paper.

Reviewer #2:This manuscript is very creative and thought-provoking because it makes me think more about very steep positive density-dependent gradients in a new light, as facilitating coexistence rather than being a unique part of species' ecology. For example, bighorn sheep suffer strong Allee effects because small groups are depredated quickly by mountain lions. This is a terrific example of the Allee effect, but one rarely thinks of this sort of positive density dependence as being a mechanism of species coexistence between large ungulates. This paper makes me think about that.I do think that the model needs more contextualization in biology from top to bottom. I had to read very carefully, and twice, to understand what I did – for eLife, the paper has to sing so that readers get most of it on the first read. There is an opportunity, and I would argue a necessity, to clarify intra- and interspecific density-dependent processes in the Introduction. The choice of terms in the modeling equations themselves could also be better tied to biology to explain exactly what kinds of systems they represent, and how broadly they can be interpreted. My specific comments are below:

I thank reviewer #2 for this positive feedback; I am glad that my study made him/her think about density dependence and coexistence in a new light. In the revised manuscript, I aimed at making this theoretical study more easily accessible by using some “real-life” examples. I also followed the reviewer #1’s suggestion to add a Glossary defining terms describing species interactions, density dependence and frequency dependence. I carefully used those terms to describe the modeling equations, making them better tied to the biological processes. I hope that by doing so, I addressed the reviewer #2’s concerns successfully.

1) Negative density dependence favors coexistence if intraspecific density dependence is stronger than interspecific density dependence. Positive density-dependence seems like a strange mechanism to promote species coexistence when it is interspecific because the most abundant species should only become a stronger and stronger competitor as it rises in abundance. Intraspecific positive density dependence – e.g. Allee effects – seems to make it more difficult for rare species to invade, so it does not promote global stability (subsection “Analytical resolution”, second paragraph). As I understand the analytical part of the model (and I have to be honest, I only read Supplementary file 1A), positive density dependence is only intraspecific because the D(ni) function for each species only includes individuals of that species. s it is always the same between species, though so their density dependence varies perfectly in tandem. I think this is really what struck me as counterintuitive because if I were to think about positive density dependence in the context of competition, I would imagine it to operate interspecifically as well so that the more abundant a species was, the better it would be able to outcompete its competitors (45-47). If I am right about what the model means, then it might be very helpful in the Introduction to explain not just "positive density dependence" writ large, but the different ways it might function (e.g. interspecifically versus intraspecifically, or using specific natural history examples that the model applies to and contrasting them to natural history examples where it does not). The fourth paragraph of the Introduction starts to do this, but are very abstract and don't go into enough detail about differences in kinds of positive density dependence, only its general effects.

By using the terms defined in the Glossary (Box 1), I clarified the nature of each density-dependent process, including conspecific positive density dependence. In particular, as correctly explained by the reviewer #2, I modeled conspecific positive density dependence by including only the density of the focal species in the term D(ni). Nonetheless, this does not mean that species densities “varies perfectly in tandem” through this mortality term. Indeed, if one species is more abundant than the other, then mortality rates are different (because species densities are different). Conspecific positive density dependence is not interspecific but it can affect species coexistence by changing the densities determining the strengths of competition or reproductive interference. I hope this is now clarified.

Following the suggestions of reviewer #1, I also tried to clarify the nature of conspecific positive density dependence by using a comparative table (Figure 1) and specific natural history examples (Figure 2) (Introduction).

2) At very high values of positive frequency dependence, the effects of positive density dependence essentially disappear because the populations hit zero mortality so quickly. I could imagine this being very important in cases where there are for example two aposematic species and both have sufficiently high population sizes that both are protected from predation. Perhaps it would also make sense to imagine two species of pack-hunting carnivores that have zero success alone, but much greater success with even a few individuals, so the two species might be able to coexist as long as both are already present in sufficient abundance. This is sort of an Allee effect too, of course, which are also experienced by species of herd animals that cannot survive in small groups, because they are too vulnerable to predation. But this model seems to describe species that are very similar in their ecology, not just because they are forced to be so to analyze the effects of density dependence, but also because density has identical intraspecific effects. Is this true? If so it would be helpful to explain in the text, and if not, then it would be helpful to explain how broadly it applies.

I thank reviewer #2 for this succinct and accurate summary of the reason why conspecific positive density dependence associated with reduced mortality can favor coexistence. Reviewer #2 is also right about the modeling assumption; species have identical intraspecific effects (this is now clarified in manuscript; subsection “Two-species model with asymmetric competition for resources”). But one must remember that even though the functions describing intraspecific effects are identical in the different species, if species differ in their density (e.g. via interspecific effects) then their changes in density due to intraspecific effects differ. Therefore, the model exactly consider the situation described by the reviewer at the beginning of this comment.

Following one of the reviewer #3’s suggestions, however, I relaxed this assumption by conducting simulations where only a fraction of species incur conspecific positive density dependence, and I showed that the outcome is qualitatively similar (Figure 7—figure supplement 8; subsection “Numerical simulations”).

3) In (2), has the carrying capacity – or effective carrying capacity – also changed? Is the purpose of this model to contrast with (1), where one spp. is a superior competitor, whereas here in (2), one spp. is a superior reproducer/survivor?

Exactly, one species is a superior survivor in model (2) and one species is a superior reproducer in model (3); and the effective carrying capacity – density values reached at equilibrium without competitor – remains the same for the two species (just like all other assumptions made in the first model). I clarified this last point (subsection “Two-species models with other forms of competitive exclusion”).

4) For Equation 3, I need more hand-holding to explain what's going on. It looks like the new term n2/(n2 + a'1*n1) means the frequency of individuals in the community that is composed of n2 – is that correct? So is this a way of incorporating positive interspecific frequency dependence into your model that already includes positive intraspecific density dependence? Would there be any difference if you modeled reproductive interference as an effect on d rather than on ni, since in a literal way it should affect reproduction rather than carrying capacity? I could also imagine it not matter if its influence is only supposed to be manifest by decreasing dn2/dt, though.

In the revised manuscript, the term of reproductive interference is now transformed in a way that make it clearer (though this is mathematically the exact same term) because it make it explicit that the frequency of heterospecifics (n1/n2) reduces the intrinsic growth rate (heterospecific negative frequency dependence, Box 1). I also present this mathematical term more thoroughly in the text (subsection “Two-species models with other forms of competitive exclusion”).

Given that reproductive interference reduces the intrinsic growth rate (e.g., the first term within the brackets = 1 in species 1 is not affected by reproductive interference; this is now clearer in the revised form of the equation), we cannot model this process as a change in the term of mortality rate.

Reviewer #3:Positive density-dependence is widely observed in nature. It is widely believed that positive density-dependence inhibits species coexistence. The main result of this manuscript is that a positive density-dependence on mortality, under certain forms of population dynamics, increases the feasibility domain of species coexistence. The paper is well-written (but I am not sure if the math here would be easily accessible to the majority of the readership at eLife). I applaud the author's efforts in proposing potentially new mechanisms of species coexistence by positive density-dependence, which could be an important contribution to the coexistence theory. However, the author needs to make more efforts in building biological motivation and checking the robustness of the results.

I thank reviewer #3 for this succinct (and accurate) summary and for this positive feedback. In the revised manuscript, I made considerate effort in building biological motivation (by using natural history examples), in checking the robustness of the results (with more than 10 new supplementary figures), and in offering guidelines for testing empirically my theoretical predictions. I hope that I addressed the reviewer’s concerns successfully.

1) The biological relevance.– The author motivated the paper by pointing up the ubiquitousness of positive density-dependence (Introduction, fifth paragraph). However, if I understood correctly, most references focus on the Allee effect of the birth rates. Then, is there substantive literature of empirical evidence that positive density-dependence acts on mortality? To be clear, I am not saying that there is not, but to encourage the author to be more explicit about the biological motivations.

Following one of reviewer #1’s suggestions, I added a comparative table to describe the different types of conspecific positive density dependence (Figure 1), and I included some natural history examples (Figure 2). While there are many putative cases of such density dependence, whether this form of density dependence is pervasive in ecological communities remains an open question. This is now discussed in the Discussion. In particular, I emphasize that this form of positive density dependence may take place even if we observe an overall pattern of conspecific negative density dependence (as in many plant communities). Even if this is not the dominant density dependent force, it can still promote species coexistence as predicted by the theoretical models.

– In the fifth paragraph of the Discussion, the author concluded that it is yet impossible to validate the results. I apologize if I misunderstood, but I feel that the paper offers no guidelines for empirical tests. To be clear, I am not asking the author to run analysis on some empirical data, but to encourage the author to discuss how this theory can be potentially tested. Otherwise, it may leave the reader the impression of a math exercise disconnected from ecological nature.

Following the reviewer’s suggestion, I now offer guidelines for assessing the prevalence of conspecific positive density dependence and for testing empirically its implication for coexistence (Discussion). Given that I got the most striking results in the case with asymmetric reproductive interference, I encourage empiricists working on organisms undergoing reproductive interference to test the implication of conspecific positive density dependence on coexistence.

2) The robustness of the results. I understand the following requests can be a bit much, but I sincerely believe that when one proposes a new ecological mechanism, the author needs to prove that it is theoretically robust to convince the empiricists. However, I am happy for a discussion if the author disagrees with anything follows.

By running new simulations based on the reviewer’s suggestions, I confirmed that my results are robust to changes in modelling assumption. I detail each of those supplementary analyses below.

– This paper only studies one particular functional form of mortality with positive density-dependence. As the author stated that "assessing the functional form of positive density-dependence in natural populations is tedious", I think the author needs to study other functional forms of positive density-dependence, at least in the 2-species case.

This is a very good point. To check the robustness of my predictions to the functional form of density dependence, I conducted simulations with other linear or non-linear mortality function (represented in Figure 3—figure supplement 1; note that I could not test γ values >3; because the numerical method was not reliable in that condition). I got similar results as in the main analyses in the two-species models (subsection “Analytical resolution”; subsection “Numerical resolutions; Figure 4—figure supplements 3 and 4, Figure 6—figure supplements 3-6) but also in the multispecies models (subsection “Numerical simulations”; Figure 7—figure supplements 6 and 7).

– This paper does not have a sort of null model to estimate the effects of positive density-dependence. That is, whether the effects of positive density-dependence solely comes with the nonlinear functional form or because of the positiveness. To do this, I suggest the author add some additional tests on negative density-dependence with the same functional response, to see that the effects are actually coming from positiveness.

I acknowledge that there was no sort of null model. In the revised version, however, I now run simulations where parameter s associates with reduced mortality and not density dependence. The predictions are similar as in the main analysis, highlighting that reduced mortality leads promotes species coexistence (subsection “Numerical resolutions”; Figure 6—figure supplements 3-4), as emphasized in the Abstract and the Introduction.

– This paper considers the same functional form of positive density-dependence for every species. However, I think it is worth examining some with positive density-dependence and some without in multi-species case.

I conducted simulations with the multispecies model (accounting for random communities) where only a fraction of species undergo conspecific density dependence. I get qualitatively the same result (subsection “Numerical simulations”; Figure 7—figure supplement 8).

– To be honest, I am a bit lost why section 1 exists. I thought the author tries to prove that positive density-dependence helps coexistence, but why spent so much time on a particular case where positive density-dependence inhibits coexistence?

I still believe that this is necessary to start the study by the analysis of this model. The first model based on Lotka equations in their purest form is analyzed analytically, precisely pinpointing the effect of the positive density-dependence factor on coexistence. While positive density-dependence does not promote coexistence in this case, we observe a recover of the coexistence equilibrium via reduced mortality, and this is exactly how conspecific positive density dependence promotes coexistence in the other models. Therefore, I think that the time spent in reading this particular case then eases the reading of the two other parts (precisely because the underlying mechanism promoting the recover of the coexistence equilibrium is made clear). This is why I have not changed the structure of the paper in the revised manuscript; I hope the reviewer appreciates my point.

– I don't find the parametrization in Supplementary file 1D satisfactory. The range of parameters seems very arbitrary to me. I was puzzled that why no other parametrizations were examined.

I conducted simulations with other range of parameters. I get qualitatively the same result(subsection “Numerical simulations”; Figure 7—figure supplement 1).